# COVID-19 among workers of a comprehensive cancer centre between first and second epidemic waves (2020): a seroprevalence study in Catalonia, Spain

Paula Peremiquel-Trillas,[1,2,3,4] Anna Saura-Lázaro [1,2]
Yolanda Benavente-Moreno [1,2,3] Delphine Casabonne [1,2,3] Eva Loureiro,[5,6]
Sandra Cabrera [7] Angela Duran,[8] Lidia Garrote,[9] Immaculada Brao,[10]
Jordi Trelis,[4,11] Maica Galán,[12] Francesc Soler,[13] Joaquim Julià,[14,15]
Dolça Cortasa,[16] Maria Ángeles Domínguez,[17,18,19] Adaia Albasanz-Puig [20,21,22]
Carlota Gudiol [20,21,22] Dolors Ramírez-Tarruella,[23] Joan Muniesa,[5]
Juan Pedro Rivas,[5] Carles Muñoz-Montplet [5,24,25] Ana Sedano,[26]
Àngel Plans [27] Beatriz Calvo-Cerrada,[27] Candela Calle,[28] Ana Clopés,[29]
Dolors Carnicer-Pont [2,30,31,32] Laia Alemany [1,2,3]
Esteve Fernández [2,4,30,31,32]

PP-T and AS-L contributed equally.

DC-P, LA and EF are joint senior authors.

For numbered affiliations see end of article.

**Correspondence to**
Dr Esteve Fernández;
efernandez@iconcologia.net

## ABSTRACT

**Objectives** Patients with cancer are at higher risk for severe COVID-19 infection. COVID-19 surveillance of workers in oncological centres is crucial to assess infection burden and prevent transmission. We estimate the SARS-CoV-2 seroprevalence among healthcare workers (HCWs) of a comprehensive cancer centre in Catalonia, Spain, and analyse its association with sociodemographic characteristics, exposure factors and behaviours.

**Design** Cross-sectional study (21 May 2020–26 June 2020).

**Setting** A comprehensive cancer centre (Institut Català d'Oncologia) in Catalonia, Spain.

**Participants** All HCWs (N=1969) were invited to complete an online self-administered epidemiological survey and provide a blood sample for SARS-CoV-2 antibodies detection.

**Primary outcome measure** Prevalence (%) and 95% CIs of seropositivity together with adjusted prevalence ratios (aPR) and 95% CI were estimated.

**Results** A total of 1266 HCWs filled the survey (participation rate: 64.0%) and 1238 underwent serological testing (97.8%). The median age was 43.7 years (p25–p75: 34.8–51.0 years), 76.0% were female, 52.0% were nursing or medical staff and 79.0% worked on-site during the pandemic period. SARS-CoV-2 seroprevalence was 8.9% (95% CI 7.44% to 10.63%), with no differences by age and sex. No significant differences in terms of seroprevalence were observed between onsite workers and teleworkers. Seropositivity was associated with living with a person with COVID-19 (aPR 3.86, 95% CI 2.49 to 5.98). Among on-site workers, seropositive participants were twofold more likely to be nursing or medical staff.

### Strengths and limitations of this study

► Seroepidemiological study with a large sample size settled in a comprehensive cancer centre.
► Questionnaire completeness was very high, with no variables presenting more than 5% of missing values.
► Recall bias is possible as the data for the correlates of SARS-CoV-2 infection rely on a self-administered questionnaire.
► The accomplishment of preventive measures might be overestimated: response and perception biases must be considered, as well as complacency bias.
► Answers reported in the questionnaire could be influenced by the participants' knowledge regarding their COVID-19 status.

Nursing and medical staff working in a COVID-19 area showed a higher seroprevalence than other staff (aPR 2.45, 95% CI 1.08 to 5.52).

**Conclusions** At the end of the first wave of the pandemic in Spain, SARS-CoV-2 seroprevalence among Institut Català d'Oncologia HCW was lower than the reported in other Spanish hospitals. The main risk factors were sharing household with infected people and contact with COVID-19 patients and colleagues. Strengthening preventive measures and health education among HCW is fundamental.

## INTRODUCTION

Front-line healthcare workers (HCWs) dealing with COVID-19 have higher exposure

to SARS-CoV-2 than the general population,[1] and they can contribute to the spread of COVID-19 as per their exposure to vulnerable patients. Since the beginning of the pandemic, several studies have been published on SARS-CoV-2 infections prevalence in HCW, although with diverse results. A meta-analysis of 49 studies, including 127 480 HCWs, showed that the overall seroprevalence of SARS-CoV-2 antibodies in the European region was 8.5%.[2] HCW in Spain have been highly affected: a total amount of 154 636 cases among HCWs were already officially notified by 2 December 2021 at the onset of the sixth pandemic wave.[3 4]

Patients with cancer are vulnerable, presenting a high risk for COVID-19 infection and more severe outcomes due to their immunosuppression status.[5] The pandemic has presented unprecedented professional and personal challenges for the oncology community.[6] Data are lacking on the seroprevalence of SARS-CoV-2 among HCW in oncological centres, and small sample sizes limit the few published studies. This study aims to estimate the seroprevalence of SARS-CoV-2 and associated sociodemographic and behavioural risk factors among workers of the Catalan Institute of Oncology (ICO), a Comprehensive Cancer Centre comprised of four hospitals in Catalonia (Spain), covering around 40% of the adult population in Catalonia.[7]

## PARTICIPANTS AND METHODS
### Study design and setting
A cross-sectional study including blood sample collection and a self-administered questionnaire was conducted between 21 May 2020 and 26 June 2020 in the four ICO centres (L'Hospitalet de Llobregat, Badalona, Tarragona/Terres de l'Ebre and Girona).

The study population were HCW delivering care and services to patients (directly or indirectly) and support staff, including those who do not deliver care but work in other tasks within the hospital. A total of 1969 employees of ICO were invited to participate in the study through an email that allowed access to the study information. The inclusion criteria were: (1) to be an active worker during the epidemic period, (1 February 2020–26 June 2020) and (2) to be aged ≥18 years. The participants filled in an online epidemiological questionnaire and were scheduled for serology testing by the Occupational Health Department. A total of 1266 HCW filled in the online epidemiological questionnaire (participation rate: 64.3%) and 1238 of them (97.8%) underwent a serology test. Three participants with inconclusive serological results were excluded. The final analysis included 1235 participants (figure 1).

### Epidemiological questionnaire and study variables
An epidemiological questionnaire was programmed online to collect information regarding sociodemographic characteristics, working information, compliance of personal protective equipment (PPE) measures

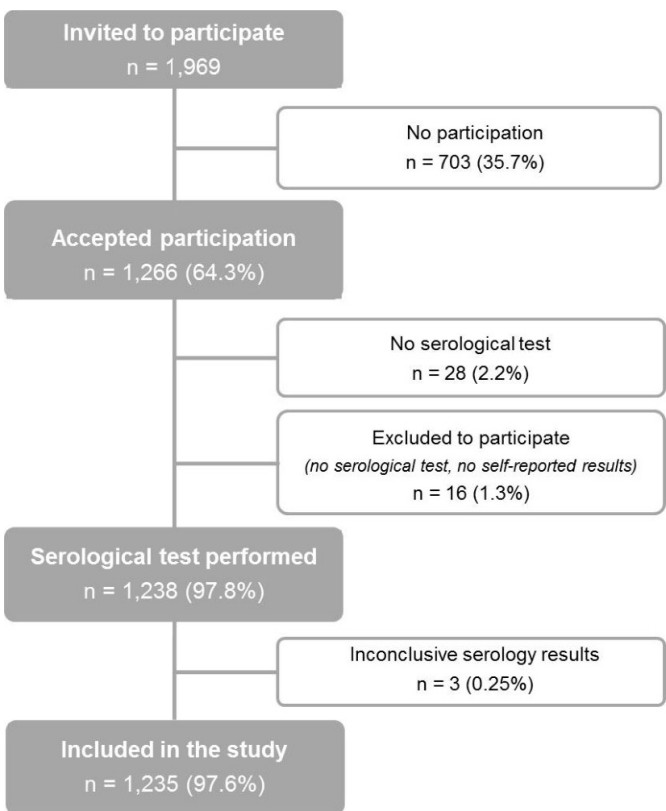

**Figure 1** Participants' flow chart in the seroprevalence survey, Catalan Institute of Oncology. 21 May 2020–26 June 2020; Spain.

at work, at home and history of previous COVID-19 infection (online supplemental material). The questionnaire was developed based on previous epidemiological studies conducted within the ICO centres, and a modified version was used in another seroprevalence study performed among university personnel of the University of Barcelona.[8]

Sociodemographic characteristics included information on age and sex, ICO centre of recruitment, presence of comorbidities, smoking history, pregnancy and cohabitants.

Work-related conditions included the professional category, teleworking status, type of shift, working on a COVID-19 area, contact with COVID-19 cases, contact with biological samples and reporting to be exposed to COVID-19.

Concerning PPE measures at work, participants were asked about feeling protected with PPE and compliance with PPE measures. Regarding the application of preventive measures outside the working setting, participants were asked if they got a shower after leaving the workplace or when arriving home, if they changed clothes after work or on home arrival, as well as about hand washing and use of face mask when shopping. Information about COVID-19 cases and protective measures were also collected among those participants reporting cohabitants. Participants were also asked about the type of transport used to go to work.

Participants were asked about a previous diagnosis of COVID-19 performed by rRT-PCR or serology test and date of diagnosis, as well as reporting COVID-19 compatible symptoms, and the type of symptoms.

## SARS-CoV-2 laboratory testing

Serum samples from participants at L'Hospitalet, Girona and Tarragona/Terres de l'Ebre were studied at the Microbiology Department of Hospital de Bellvitge and samples from HCWs at ICO Badalona were analysed at the MetroNord Regional Clinical Laboratory, using the same procedures and techniques in both laboratories. Detection of SARS-CoV-2 antibodies was carried out using the quantitative SARS-CoV-2 S1/S2 IgG LIAISON test (DiaSorin, Vercelli, Italy) on the LIAISON XL platform, following the manufacturer's instructions. This test discriminates among negative (<12AU/mL; with 3.8 as IgG detection limit), equivocal (12.0–15.0AU/mL) and positive (>15.0AU/mL) subjects. In those cases in which (1) IgG anti S1/S2 quantification was higher than the limit of detection (ie, >3.8AU/mL) but did not reach the limit of discrimination (ie,<15AU/mL) and/or (2) when the HCW answered the questionnaire saying that he or she had been diagnosed of COVID-19 but IgG anti S1/S2 where lower than 15 AU/mL, an additional serological study was performed using a different antigen (N) as a target. In this case, a SARS-CoV-2 IgG test (Abbott Diagnostics, Sligo, Ireland) was run on an Architect i2000 platform. This test discriminates among negative (<1.4Index (S/C)) and positive (≥1.4Index (S/C)) subjects.

## Case definition

A seropositive case of SARS-CoV-2 was defined as seropositivity to IgG independently of previous self-reported results.

## Patient and public involvement

No patient was involved in the study.

## Statistical analysis

Crude global and by subgroups SARS-CoV-2 seroprevalence and 95% CIs were calculated. Differences in the distribution of study variables between seropositive and seronegative participants were assessed using $\chi^2$ test for categorical variables, and parametric or non-parametric tests were performed for normal and non-normal continuous variables, respectively. Prevalence ratios (PR) and 95% CIs were estimated using Poisson regression models with robust variance.[9] Adjusted PRs (aPRs) were used for statistically significant variables in the bivariate analysis and those considered relevant for the study design. Thus, adjusted models included sex, ICO centre of recruitment, age, type of HCW, teleworking and cohabitants. Linear trends for variables with ordinal categories was based in the likelihood ratio test of the model with the ordinal variable as a continuous one. P values were based on two-sided hypothesis tests and considered significant at p<0.05. All analyses were conducted using Stata V.16.0 (StataCorp).

## RESULTS

A total of 1235 HCWs with serological results (figure 1) were included in the analysis: 76.0% were female, the median age was 43.7 years (p25–p75: 34.8–51.0 years), 52.2% were nursing or medical staff and 18.6% of the participants teleworked full time during the study period (table 1). Up to 14.7% of the participants reported at least one comorbidity. Regarding smoking habits, 16.0% were current smokers and 28.2% reported to be former smokers (table 1). Seven women were pregnant, and none of them showed seropositivity.

The overall crude SARS-CoV-2 seroprevalence was 8.9% (95% CI 7.44 to 10.63%), with no statistically significant differences by neither age group nor sex, and the seroprevalence for nursing and medical staff was 11.6% (95% CI 9.37 to 14.34%). After fully adjustment, the main determinants of higher seroprevalence included working at ICO Girona compared with workers at ICO L'Hospitalet (aPR 1.52, 95% CI 0.97 to 2.38) and nursing or medical staff compared with other groups (aPR 2.04, 95% CI 1.33 to 3.14) (table 1).

Seroprevalence among on-site workers was 8.8% (95% CI 7.15% to 10.71%) (table 2). Onsite workers were younger, assisting HCWs and reported more frequently rRT-PCR previous to serology than teleworkers, but no differences were observed in sex, self-reported comorbidities, smoking history, cohabiting with COVID-19 positive case between them and teleworkers (online supplemental material). Among this group (N=981) of professionals who never or occasionally teleworked SARS-CoV-2 seropositivity was not associated with not working in a COVID-19 area (aPR 1.29, 95% CI 0.81 to 2.06), nor being in contact with COVID-19 biological samples (aPR 1.30, 95% CI 0.77 to 2.20) nor being in contact with patients with COVID-19 (aPR 1.09, 95% CI 0.66 to 1.79) were associated with SARS-CoV-2 positivity (table 2). On-site nursing or medical staff who worked in a COVID-19 area had twofold SARS-CoV-2 seroprevalence than others who did not work in COVID-19 area (aPR 2.45, 95% CI 1.08 to 5.52). Seropositivity was higher among those whom referred being exposed by interacting with colleagues (aPR 3.26, 95% CI 1.49 to 7.15). On-site workers who self-reported symptoms of COVID-19 were almost 10-fold more likely to be seropositive than those who did not (aPR: 9.5, 95% CI 5.34 to 17.03). Most of the on-site workers were highly adherent to the recommendation of hand hygiene at work. Hand washing before eating or working, were followed by more than 97% of on-site workers, whereas around 24% of them reported not hand hygiene after working or a low frequency of handwashing during the workday. In relation to protective measures at work, 17.4% of the on-site workers did not feel protected with PPE, and 12.1% did not use PPE with confirmed or suspicious COVID-19 cases. About colleagues' behaviour, 2 m safety distance from colleagues when having lunch was reported to be unfollowed by 14.1% (table 2).

Concerning the correlates of seropositivity according to household factors for all participants (table 3),

**Table 1** Sociodemographic characteristics associated with SARS-CoV-2 positive serology among study participants (N=1235)

| | Total participants n (%) | SARS-CoV-2 seroprevalence n (%) | Prevalence (95% CI) | P value* | aPR (95% CI)‡‡ |
|---|---|---|---|---|---|
| Study participants | 1235 | 110 | 8.91 (7.44 to 10.63) | | |
| **Sex** | | | | | |
| Male | 291 (23.6) | 27 (24.5) | 9.28 (6.44 to 13.20) | | REF |
| Female | 939 (76.0) | 83 (75.5) | 8.84 (7.18 to 10.83) | 0.82 | 0.82 (0.53 to 1.28) |
| **Age (median, (p25–p75))** | 43.7 (34.8–51.0) | 42.8 (32.0–50.1) | | 0.62 | 0.99 (0.97 to 1.01) |
| <35 years | 313 (25.3) | 33 (30.0) | 10.54 (7.59 to 14.46) | | REF |
| 35–49 years | 566 (45.8) | 47 (42.7) | 8.30 (6.29 to 10.88) | | 0.85 (0.55 to 1.34) |
| >49 years | 356 (28.8) | 30 (27.3) | 8.43 (5.95 to 11.80) | 0.5 | 0.88 (0.53 to 1.46) |
| **ICO centre** | | | | | |
| ICO L'Hospitalet | 885 (71.7) | 73 (66.4) | 8.25 (6.61 to 10.25) | | REF |
| ICO Girona | 204 (16.5) | 29 (26.4) | 14.22 (10.06 to 19.72) | | 1.52 (0.97 to 2.38) |
| ICO Badalona | 134 (10.9) | 7 (6.4) | 5.22 (2.51 to 10.56) | | 0.54 (0.25 to 1.19) |
| ICO Tarragona/Terres de l'Ebre | 12 (1.0) | 1 (0.9) | 8.33 (1.16 to 41.38) | 0.02 | 1.07 (0.15 to 7.83) |
| **Professional category** | | | | | |
| Nursing staff‡ | 380 (30.8) | 43 (39.0) | 11.32 (8.50 to 14.92) | | REF |
| Medical staff§ | 265 (21.5) | 32 (29.1) | 12.08 (8.67 to 16.58) | | 1.07 (0.65 to 1.76) |
| Middle and superior technicians | 285 (23.1) | 14 (12.7) | 4.91 (2.93 to 8.13) | | 0.41 (0.22 to 0.77) |
| Service staff¶ | 114 (9.2) | 2 (1.8) | 7.02 (3.55 to 13.42) | | 0.69 (0.31 to 1.54) |
| Porter | 21 (1.7) | 8 (7.3) | 9.52 (2.39 to 31.16) | | 0.74 (0.17 to 3.24) |
| Administrative | 129 (10.4) | 8 (7.3) | 6.20 (3.13 to 11.92) | | 0.54 (0.25 to 1.16) |
| Other | 20 (1.6) | 1 (0.9) | 5.00 (0.70 to 28.26) | 0.03 | 0.50 (0.07 to 3.71) |
| Nursing or medical staff** | 645 (52.2) | 75 (68.2) | 11.63 (9.37 to 14.34) | <0.001 | 2.04 (1.33 to 3.14) |
| Other staff†† | 569 (46.1) | 33 (30.0) | 5.80 (4.15 to 8.05) | | REF |
| **Telework** | | | | | |
| Never/occasionally | 981 (79.4) | 86 (78.1) | 8.77 (7.15 to 10.71) | | REF |
| Always | 230 (18.6) | 23 (20.9) | 10.00 (6.72 to 14.63) | 0.56 | 1.60 (0.98 to 2.59) |
| **Shift work** | | | | | |
| Morning | 545 (44.1) | 49 (45.0) | 8.99 (6.86 to 11.7) | | REF |
| Evening | 140 (11.3) | 10 (9.1) | 7.14 (3.88 to 12.77) | | 0.56 (0.34 to 0.93) |
| Split shift (morning–evening) | 417 (33.8) | 38 (34.5) | 9.11 (6.7 to 12.28) | | 0.88 (0.57 to 1.37) |
| Night | 88 (7.1) | 10 (9.1) | 11.36 (6.22 to 19.86) | | 0.95 (0.46 to 1.96) |
| Other | 25 (2) | 3 (2.7) | 12 (3.92 to 31.32) | 0.83 | 1.15 (0.35 to 3.75) |
| **Comorbidities**** | | | | | |
| None | 1054 (85.3) | 99 (90.0) | 9.39 (7.77 to 11.31) | | REF |
| Yes | 181 (14.7) | 11 (10.0) | 6.08 (3.4 to 10.64) | 0.15 | 0.67 (0.36 to 1.25) |
| **Smoking history** | | | | | |
| Never | 650 (52.6) | 80 (72.7) | 12.31 (9.99 to 15.07) | | REF |
| Past | 348 (28.2) | 22 (20.0) | 6.32 (4.20 to 9.42) | | 0.57 (0.35 to 0.93) |
| Current | 198 (16.0) | 8 (7.3) | 4.04 (2.03 to 7.87) | 0.0002 | 0.38 (0.18 to 0.79) |

Continued

**Table 1** Continued

| | Total participants | SARS-CoV-2 seroprevalence | Prevalence (95% CI) | P value* | aPR (95% CI)‡‡ |
|---|---|---|---|---|---|
| | n (%) | n (%) | | | |
| Cohabitants | | | | | |
| Yes | 1119 (90.6) | 95 (86.0) | 8.49 (6.99 to 10.27) | | REF |
| No | 104 (8.4) | 15 (13.6) | 14.42 (8.88 to 22.57) | 0.04 | 1.48 (0.83 to 2.66) |

Numbers do not always sum up the total due to some missing values (none of the categories present more than 5% of missing values).

*Comorbidities: hypertension, obesity (BMI ≥30), heart disease, liver disease, diabetes, chronic respiratory disease, renal disease, cancer, autoimmune disorders and other immunological disorders.

† χ2 test for categorical variables (Fisher's exact test corrected for continuity) and median test for continuous variables.

‡Nursing staff: nurses and nursing assistants.

§Medical staff: resident physicians and specialists.

¶Service staff: security, maintenance, cleaning and kitchen.

**Nurses, nursing assistants, resident physicians and specialists.

††Middle and superior technicians, security, maintenance, cleaning, kitchen, porter, administrative and other.

‡‡ Adjusted for sex, age (continous), ICO centre, telework and cohabitants.

aPR, adjusted prevalence ratio; BMI, body mass index; ICO, Institute of Oncology ; p25, 25% percentile; p75, 75% percentile.

seropositivity was associated with living with a COVID-19 positive person (aPR 3.86, 95% CI 2.49 to 5.98). Up to 17.3% of the participants did not take a shower nor change clothes on home arrival, but the majority (99.0%) did hand hygiene. The least followed hand hygiene home practices were after money, phone and other personal tools manipulation and after nose blowing, coughing or sneezing (23.5% and 22.7%). However, not following protection measures or hand hygiene at home were associated with a higher SARS-CoV-2 seroprevalence.

Clinical characteristics were collected for those participants (N=469) who reported a rRT-PCR performed previous to serology (online supplemental material). The majority of the patients with a positive serology and reporting a positive rRT-PCR presented compatible COVID-19 symptoms (74.4%). Among seropositive patients, the most common symptoms were arthromyalgia, cough, headache, asthenia and anosmia. Reporting a positive rRT-PCR when presenting compatible symptoms was associated with a threefold higher prevalence of seropositivity (aPR 3.10, 95% CI 1.78 to 5.31). An increased number of compatible symptoms was also associated with a higher seroprevalence (aPR 7.4, 95% CI 1.78 to 5.31, for presenting four or more symptoms compared with no symptoms).

## DISCUSSION

Despite the impact of COVID-19 in oncological patients,[10] there are scarce SARS-CoV-2 seroprevalence studies in comprehensive cancer centres with large sample sizes. The global SARS-CoV-2 seroprevalence was 8.9% during the first wave of the COVID-19 pandemic, lower than expected, owing to the presumed higher risk among HCW. Also, it was lower than the reported estimates in two studies performed among HCW in Catalonia between March-April and May 2020, showing a seroprevalence of 11.2%[11] and 10.3%,[12] respectively. In all cases, the seroprevalence was higher than in the general population, estimated to be of a maximum of 7.4% in the Barcelona metropolitan area when the study was conducted.[13] Seroprevalence studies interpretation must be related to the average COVID-19 prevalence at the time of blood collection. Both of the mentioned studies were carried out earlier than ours, which was performed approximately 1 month later (21 June 2020 May–26 June 2020), and 2 months after the first-wave peak in Catalonia (23 March).[14] Another explanation for this lower seroprevalence in our Centre concerns the participation: all active HCW, regardless of their teleworking status during the previous months or work absenteeism, were invited to participate and most did (64%). In contrast, Garcia-Basteiro et al's[11] and Barallat et al's[12] studies comprised general hospitals[10 11] and primary healthcare centres[12] in which the incidence could be higher than in a monographic cancer centre.

Several studies regarding COVID-19 infections in HCW in Spain have been published, although showing diverse results. In a tertiary-care hospital in Mallorca, with low regional seroprevalence in the general population (<2%), the prevalence of infected HCW (n=2210) was 2.8%.[15] Varona et al performed a cross-sectional study evaluating 6038 employees from the healthcare system of 17 hospitals across four regions in Spain (Madrid, Catalonia, Galicia and Castilla-Leon), showing an 11% seropositivity for SARS-CoV-2 IgG.[16] Finally, other studies in Madrid reported a seroprevalence between 16.6% and 36.5% among HCW in areas with high COVID-19 prevalence.[17–19] These studies revealed seroprevalence of SARS-CoV-2 IgG antibodies in HCW tend to be higher than in the general population, at variance according to regional COVID-19 incidence.

The prevalence of SARS-CoV-2 antibodies among HCW has been increasingly investigated in many other countries showing a broad range of outcomes. So far, two systematic

**Table 2** Occupational factors associated with SARS-CoV-2 positive serology among on-site workers (N=981)

| | Total participants n (%) | SARS-CoV-2 seroprevalence n (%) | Prevalence (95% CI) | P value* | Adjusted PR (95% CI)† |
|---|---|---|---|---|---|
| On-site workers | 981 (79.4) | 86 (78.1) | 8.77 (7.15 to 10.71) | 0.56 | |
| Type of transport to work | | | | | |
| Private | 751 (76.6) | 66 (76.7) | 8.79 (6.96 to 11.04) | | REF |
| Public | 154 (15.7) | 15 (17.4) | 9.74 (5.95 to 15.54) | | 1.32 (0.74 to 2.36) |
| Private and public | 35 (3.6) | 2 (2.3) | 5.71 (1.43 to 20.19) | | 0.63 (0.15 to 2.58) |
| Walking | 37 (3.8) | 3 (3.5) | 8.11 (2.63 to 22.34) | 0.89 | 0.57 (0.14 to 2.35) |
| Working in a COVID-19 area | | | | | |
| No | 398 (40.6) | 29 (33.7) | 7.29 (5.11 to 10.29) | | REF |
| Yes | 545 (55.6) | 55 (63.9) | 10.09 (7.83 to 12.92) | 0.14 | 1.29 (0.81 to 2.06) |
| Type of and COVID-19 area‡ | | | | | |
| Non-assisting HCW and never worked in a COVID-19 area | 148 (15.1) | 7 (8.0) | 4.73 (2.27 to 9.6) | | REF |
| Non-assisting HCW and ever worked in a COVID-19 area | 230 (23.4) | 13 (15.1) | 5.65 (3.31 to 9.5) | | 1.12 (0.44 to 2.82) |
| Assisting HCW and never worked in a COVID-19 area | 244 (24.9) | 22 (25.6) | 9.02 (6.01 to 13.32) | | 1.81 (0.77 to 4.26) |
| Assisting HCW and ever worked in a COVID-19 area | 311 (31.7) | 40 (46.5) | 12.86 (9.57 to 17.07) | 0.006 | 2.45 (1.08 to 5.52) |
| p-trend | | | | | 0.26 |
| Contact with COVID-19 cases | | | | | |
| No | 333 (33.9) | 23 (26.7) | 6.91 (4.63 to 10.18) | | REF |
| Yes | 536 (54.6) | 57 (66.3) | 10.63 (8.29 to 13.54) | 0.07 | 1.30 (0.77 to 2.20) |
| Contact with COVID-19 biological samples | | | | | |
| No | 646 (65.9) | 51 (59.3) | 7.89 (6.05 to 10.24) | | REF |
| Yes | 282 (28.7) | 30 (34.9) | 10.64 (7.54 to 14.81) | 0.17 | 1.09 (0.66 to 1.79) |
| Reporting to be exposed to COVID-19 by interacting with colleagues at work | | | | | |
| No | 242 (24.7) | 66 (76.7) | 2.89 (1.38 to 5.95) | | REF |
| Yes | 608 (62.0) | 7 (8.1) | 10.86 (8.62 to 13.59) | <0.0001 | 3.26 (1.49 to 7.15) |
| Reporting COVID-19 compatible symptoms | | | | | |
| No | 623 (63.5) | 15 (17.4) | 2.41 (1.46 to 3.96) | | REF |
| Yes | 306 (31.2) | 68 (79.1) | 22.22 (17.91 to 27.23) | <0.0001 | 9.53 (5.34 to 17.03) |
| Not following protection measures at work | | | | | |
| Felt protected with PPE | 132 (17.4) | 12 (16.9) | 9.09 (5.23 to 15.34) | 0.83 | 0.98 (0.51 to 1.88) |
| Colleagues cover themselves with their elbows when sneezing/coughing | 155 (15.8) | 21 (24.4) | 13.55 (9.00 to 19.90) | 0.01 | 1.70 (1.01 to 2.87) |
| 2 m safety distance from colleagues during lunch | 127 (14.1) | 12 (15.6) | 9.45 (5.44 to 15.91) | 0.71 | 1.06 (0.56 to 1.99) |
| Use of PPE with confirmed or suspicious COVID-19 patients | 79 (12.1) | 7 (10.45) | 8.86 (4.28 to 17.46) | 0.63 | 1.01 (0.45 to 2.26) |
| PPE removal safety | 48 (7.3) | 3 (4.6) | 6.25 (2.03 to 17.68) | 0.33 | 0.54 (0.17 to 1.74) |
| Personal use of mask | 34 (3.5) | 1 (1.2) | 2.94 (0.41 to 18.17) | 0.21 | 0.41 (0.06 to 2.99) |
| Colleagues use of surgical mask | 7 (0.7) | 1 (1.2) | 14.29 (1.96 to 58.12) | 0.62 | 1.68 (0.23 to 12.29) |
| Not following hand hygiene at work | | | | | |
| ≤7 times during workday | 233 (23.8) | 15 (17.4) | 6.44 (3.92 to 10.41) | 0.13 | 0.71 (0.39 to 1.28) |
| After money, phone and other personal tools manipulation | 175 (17.8) | 16 (18.6) | 9.14 (5.67 to 14.41) | 0.89 | 1.00 (0.58 to 1.74) |

Continued

**Table 2** Continued

| | Total participants | SARS-CoV-2 seroprevalence | | | |
|---|---|---|---|---|---|
| | n (%) | n (%) | Prevalence (95% CI) | P value* | Adjusted PR (95% CI)† |
| Every time entering in a new workspace | 102 (10.4) | 5 (5.8) | 4.90 (2.05 to 11.25) | 0.14 | 0.55 (0.22 to 1.37) |
| Before working | 21 (2.1) | 3 (3.5) | 14.29 (4.67 to 36.17) | 0.37 | 1.72 (0.54 to 5.47) |
| After finishing the workday | 17 (1.7) | 1 (1.2) | 5.88 (0.82 to 32.09) | 0.67 | 0.65 (0.09 to 4.72) |
| Before eating | 9 (0.9) | 2 (2.3) | 22.22 (5.59 to 57.95) | 0.16 | 2.67 (0.65 to 10.94) |

Numbers do not always sum up the total due to some missing value (none of the categories present more than 5% of missing values).
*χ2 test.
†Adjusted for sex, age (continuous), ICO centre, care staff, telework and cohabitants.
‡Assisting HCW: nurses, nursing assistants, resident physicians and specialists; otherwise, classified and non-assisting HCW.
HCW, healthcare worker; ICO, Institute of Oncology; PPE, personal protective equipment; PR, Prevalence Ratio.

reviews estimated an overall seroprevalence of SARS-CoV-2 antibodies of 8.7% and 8.0% among 127 480 HCW and 168 200 HCW, respectively, before vaccination started.[2 20] Seroprevalence was higher in studies conducted in North America (12.7%) compared with those conducted in Europe (8.5%), Africa (8.2) and Asia (4%).[2]

In Europe, seroprevalence rates among HCW in Germany, Denmark and Belgium were low (1.6%, 4.0% and 6.4%, respectively).[21–23] These studies were conducted during early stages of the epidemic, and therefore, they derived that infection was community acquired. Also, the Belgian study, with a sample size of almost 30 000 HCW, notes that the high availability of PPE, high standards of infection prevention and PCR screening in symptomatic staff, coupled with contact tracing and quarantine, might explain the relatively low seroprevalence.[23] An study performed in Lombardy, Italy,[24]

**Table 3** Household factors associated with SARS-CoV-2 positive serology among study participants (n=1235)

| | Total participants | SARS-CoV-2 seroprevalence | | | Adjusted PR (95% CI)† |
|---|---|---|---|---|---|
| | n (%) | n (%) | Prevalence (95% CI) | P value* | |
| Study participants | 1235 | 110 | 8.91 (7.44 to 10.63) | | |
| **Cohabitants with COVID-19‡** | | | | | |
| No | 894 (79.9) | 52 (54.7) | 5.82 (4.46 to 7.56) | | REF |
| Yes | 141 (12.60) | 34 (35.8) | 24.11 (17.76 to 31.86) | <0.0001 | 3.86 (2.49 to 5.97) |
| **Cohabitants cover themselves with their elbow when sneezing** | | | | | |
| No | 158 (14.1) | 18 (18.9) | 11.39 (7.29 to 17.37) | | REF |
| Yes | 919 (82.1) | 73 (76.8) | 7.94 (6.36 to 9.88) | 0.15 | 0.73 (0.43 to 1.22) |
| **Not following protection measures at home§** | | | | | |
| Use of face mask when shopping | 17 (1.4) | 2 (1.8) | 11.76 (2.95 to 36.86) | 0.67 | 0.98 (0.24 to 4.05) |
| Shower and clothes changing afterwork or on home arrival | 214 (17.3) | 20 (18.2) | 9.35 (6.11 to 14.05) | 0.82 | 1.02 (0.62 to 1.69) |
| **Not following hand hygiene at home§** | | | | | |
| On arrival | 12 (1) | 2 (1.8) | 16.67 (4.19 to 47.76) | 0.35 | 1.59 (0.39 to 6.60) |
| Before eating | 60 (4.9) | 9 (8.2) | 15.00 (7.99 to 26.4) | 0.09 | 1.55 (0.77 to 3.12) |
| After money, phone and other personal tools manipulation | 290 (23.5) | 27 (24.6) | 9.31 (6.46 to 13.24) | 0.71 | 1.01 (0.65 to 1.58) |
| After cleaning | 110 (8.9) | 8 (7.3) | 7.27 (3.68 to 13.88) | 0.53 | 0.78 (0.38 to 1.61) |
| After nose blowing | 280 (22.7) | 25 (22.7) | 8.93 (6.1 to 12.88) | 0.99 | 0.93 (0.58 to 1.48) |

Numbers do not always sum up the total due to some missing values (none of the categories present more than 5% of missing values).
*χ2 test.
†Adjusted for sex, age (continuous), ICO centre, care staff, telework and cohabitants.
‡Analyses performed among those participants who reported having cohabitants (n=1119).
§Unfollowing the measures of protection and hand hygiene recommendations.
ICO, Institute of Oncology; PR, Prevalence Ratio.

one of the Italian regions most hit by the first epidemic wave, showed a seroprevalence of 7.4% (3.8%–11.0%), similar to the observed in the Catalan studies.[11 12] Sweden and the UK were the two European countries reporting the highest seropositivity rates among HCW: 19.1% and between 18.0% and 45.3%, respectively.[25–27] In the UK, this high seroprevalence was settled in London during the week with the highest number of new cases in the city in the first wave, with around 15% seropositivity among the general population. In the USA, the prevalence of infection among HCW was 10.7%, despite high variation, as low as 1.1% in California[28] to 13.7% in New York State.[29]

Despite SARS-CoV-2 seropositivity rate in oncological HCW has significant implications for oncological patients, scant research has been done. The only study published with a large sample size was in Tokyo, Japan, and it showed a very low seroprevalence of 0.67% among 1,190 HCW. It was performed at the end of the first wave in Japan, between the 3 August 2020 and the 30 October 2020, so this may explain the lower seroprevalence compared with our estimation. A French study performed among 663 HCW and 1011 patients with cancer, after the end of the first wave, showed also low seroprevalence both for HCW and patients (1.8% and 1.7%, respectively).[30] Other studies that have been published were based on small sample sizes and showed very variable seroprevalence rates.[22 31–35]

In our study, we found no differences in HCW seroprevalence according to sex, age and presence of comorbidities. Current or past smoking was however inversely associated to SARS-CoV-2 seroprevalence. Early studies in selected cohorts of COVID-19 patients showed a paradoxical higher risk of SARS-CoV-2 infection among non-smokers[36] while ever smokers showed higher risk of COVID-19 progression, including severity of the disease, intensive care unit admission and death.[37]

It is worth mentioning that, unlike most of the other published seroepidemiological studies among HCW, this study was performed among all the HCWs of the institution, regardless they did full-time telework during the study period (21.6%). No differences by telework were found, and among all study participants the main factor associated with SARS-CoV-2 seropositivity was living with a COVID-19 case, with a 1.5 times higher probability, similarly to what has been described in other studies.[2 20] This finding supports the importance of community dissemination of the infection also for HCWs.

Our study shows that among on-site HCW in an oncological centre, working as medical care staff (nursing, nursing assistant, resident physicians and specialists) in COVID-19 areas stood out as one of the main factors associated with developing SARS-CoV-2 antibodies. Published results regarding the possibility of in-hospital infection among HCW and transmission at work are controversial. Some studies did not find any relation between working in COVID-19 unit or professional category with seropositivity[11 24] whereas other studies reported that seroprevalence was strongly associated with patient related work.[16 21 22 25]

Contact with colleagues at work is potentially a risky situation for transmission among HCW as well as the relaxation of protective measures at the end of the working day. In our study, the on-site HCW who reported being exposed to COVID-19 by other colleagues presented an almost fourfold probability of being seropositive. Most of the HCWs declared to follow the protective measures at the workplace, and no differences in seroprevalence were found according to protective measures and hand hygiene.

Protecting HWC health is of paramount importance for reducing morbidity and mortality, reducing transmission and maintaining the health system capacity.[38] Thus, international health authorities recommend screening strategies for SARS-CoV-2 infection in exposed or high-risk HCW[39] as well as massive COVID-19 vaccination.[40]

Significant differences exist in SARS-CoV-2 testing between countries, and existing programmes focus on screening symptomatic rather than asymptomatic staff. Published studies point out the fact that screening should be performed regardless of the absence of typical symptoms for COVID-19 disease. It has been demonstrated that seroconversion can occur in HCW who have suffered no previous symptoms of SARS-CoV-2 infection[41 42] as asymptomatic transmission is very relevant in SARS-CoV-2 spread.[42 43] Thus, the approach for mass testing of both symptomatic and asymptomatic HCW could mitigate workforce depletion by unnecessary quarantine, reduce spread in atypical, mild or asymptomatic cases; and protect patients and healthcare workforce.

Among the potential limitations of the study, some recall bias is possible as the data for the correlates of SARS-CoV-2 infection rely on a self-administered questionnaire. Also, response and perception biases must be considered, as well as complacency bias. Results, especially those regarding the accomplishment of preventive measures, might be overestimated. Answers reported in the questionnaire could be influenced by the participants' knowledge regarding their COVID-19 status. However, this study is the first seroepidemiological study with such a large sample size settled in an oncological health centre. The sufficient sample size and high response rate (64.3%) are strengths of the study, although information regarding non-participants was not collected, and we cannot disregard a potential participation bias. However, the distribution by age and sex was similar between participants and non-participants and a possible reason for no participation is that professionals from ICO-Badalona had previously participated in an HCW county seroprevalence survey.[12] Also, the fact that the information of the study and the questionnaire was published online and sent by email, as well as the short period of time stablished to respond to it, could have limited the participation. Questionnaire completeness was very high, with no variables presenting more than 5% of missing values.

In conclusion, SARS-CoV-2 seroprevalence among ICO HCW at the end of the first wave of the pandemic was lower than the reported in other Catalan hospitals, but higher than among the general population living in the area. Whereas the main risk factor was living with infected people, among on-site workers, contact with colleagues was associated with SARS-CoV-2 infection. Knowing the seroprevalence rate and follow-up evaluation of persistence may help hospitals to characterise the staff at risk, rationalise their placement,

prioritise the use of PPE, thereby potentially reducing the risk of infection. Follow-up studies to evaluate long-term durability of antibodies among HCW will be of interest, after the introduction of COVID-19 vaccination among HCW, to better promote infection control in this group. Strengthening preventive measures and health education among HCW is fundamental, especially in oncological departments and centres.

**Author affiliations**
[1]Cancer Epidemiology Research Programme, Cancer Epidemiology and Prevention Department, Institut Català d'Oncologia (ICO), L'Hospitalet de Llobregat, Spain
[2]Epidemiology and Public Health Programme, Institut d'Investigació Biomèdica de Bellvitge (IDIBELL), L'Hospitalet de Llobregat, Spain
[3]CIBER of Epidemiology and Public Health (CIBERESP), Madrid, Spain
[4]School of Medicine and Clinical Sciences, Universitat de Barcelona, L'Hospitalet de Llobregat, Spain
[5]Computer Science Services, Technology & Physics, Institut Català d'Oncologia (ICO), L'Hospitalet de Llobregat, Spain
[6]Computational Science and Artificial Intelligence, Schoolof Computer Science of Coruña, University of Coruña (UDC), Coruña, Spain
[7]Research Nursing Department, Institut Català d'Oncologia (ICO), Badalona, Spain
[8]Nursing Department, Institut Català d'Oncologia (ICO), L'Hospitalet de Llobregat, Spain
[9]Nursing Department, Institut Català d'Oncologia, Badalona, Spain
[10]Nursing Department, Institut Català d'Oncologia (ICO), Girona, Spain
[11]Palliative Care Department and Medical Director, Institut Català d'Oncologia (ICO), L'Hospitalet de Llobregat, Spain
[12]Esofagogastric Tumours Functional Unit and Medical Director, Institut Català d'Oncologia (ICO), L'Hospitalet de Llobregat, Spain
[13]Pharmacy Service and Medical Director, Institut Català d'Oncologia (ICO), Girona, Spain
[14]Palliative Care Department and Medical Director, Institut Català d'Oncologia (ICO), Badalona, Spain
[15]School of Medicine and Health Sciences, Universitat Internacional de Catalunya, Barcelona, Spain
[16]Medical Director, Institut català d'Oncologia, Tarragona, Spain
[17]Microbiology Department, Hospital Universitari de Bellvitge, L'Hospitalet de Llobregat, Spain
[18]Infectious Diseases Programme, Institut d'Investigació Biomèdica de Bellvitge (IDIBELL), L'Hospitalet de Llobregat, Spain
[19]Department of Pathology Experimental Therapeutics, Universitat de Barcelona, L'Hositalet de Llobregat, Spain
[20]Infectious Disease Department, Hospital Universitari de Bellvitge, L'Hospitalet de Llobregat, Spain
[21]Infectious Disease Unit, Institut català d'Oncologia (ICO), L'Hospitalet de Llobregat, Spain
[22]CIBER of Infectious Diseases (CIBERINFEC), Madrid, Spain
[23]Preventive Medicine Unit, Institut Català d'Oncologia (ICO), L'Hospitalet de Llobregat, Spain
[24]Medical Physics and Radiation Protection Department, Institut Català d'Oncologia (ICO), Girona, Spain
[25]Department of Medical Sciences, Universitat de Girona, Girona, Spain
[26]Human Resources Department, Institut Català d'Oncologia (ICO), L'Hospitalet de Llobregat, Spain
[27]Occupational Health Unit, Institut Català d'Oncologia (ICO), L'Hospitalet de Llobregat, Spain
[28]General Direction, Institut Català d'Oncologia (ICO), L'Hospìtalet de Llobregat, Spain
[29]Scientific Direction, Institut Català d'Oncologia (ICO), L'Hospitalet de Llobregat, Spain
[30]Cancer Prevention and Control Programme, Cancer Epidemiology and Prevention Department, Institut català d'Oncologia (ICO), L'Hospitalet de Llobregat, Spain
[31]CIBER of Respiratory Diseases (CIBERES), Madrid, Spain
[32]WHO Collaborating Center for Tobacco Control, Institut català d'Oncologia (ICO), L'Hospitalet de Llobregat, Spain

**Acknowledgements** The authors acknowledge all the healthcare workers who participated in the study as well as all the ICO staff involved in the logistics of the different aspects of the study. We thank CERCA Programme/Generalitat de Catalunya for institutional support.

**Contributors** EF, DC-P, AP, CC, AC and AS-L contributed to study design. SC, AD, LG, IB, JT, MG, FS, JJ, DC, BC-C, DR-T, CG, AA-P and AP accrued participants and care for blood collection at ICO centres. Laboratory analyses were coordinated by MADL. The questionnaire was designed by DC-P and EF, and revised by PP-T, AS-L, YBM, DC, AP and LA. Questionnaire's implementation was done by EL, JM, JPR and CM-M. Data were analysed by YB and DC. PP-T, AS-L, YB, DC, LA, and EF interpreted the initial results and designed the tables. All authors contributed to interpretation of results. The first draft of the manuscript was prepared by PP-T and AS-L and EF. PP-T, AS-L, YB, DC, LA, DC-P and EF were the main contributors to the writing of the manuscript. All authors assisted in manuscript review. The co-senior authors had full access to all the data in the study for interpretation and had final responsibility for manuscript generation and review, and the decision to submit for publication. EF is the guarantor.

**Funding** This work was supported by the Catalan Institute of Oncology. PP-T is partially supported by the Instituto de Salud Carlos III (ISCIII), Government of the Kingdom of Spain, cofunded by FEDER funds/European Regional Development Fund (ERDF)-a way to build Europe (CM19/00216). The Ministry of Universities and Research, Government of Catalonia partly supports EF and DCP (2017SGR319) and PP-T, AS-L, YBM, DC and LA (2017SGR1085).

**Competing interests** JJ received research funding from Kyowa Kirin and from Angelini Pharma for congress attendance. The Cancer Epidemiology Research Programme has received grants from Merck & Co., Roche, GSK, IDT, Hologic and Seegene.

**Patient and public involvement** Patients and/or the public were not involved in the design, or conduct, or reporting, or dissemination plans of this research.

**Patient consent for publication** Not applicable.

**Ethics approval** The present study was approved by the Hospital Universitari de Bellvitge Ethics Committee (PR205/20). The study follows the Helsinki Declaration and subsequent amendments, and Spanish data confidentiality laws (General data protection regulation Organic Law 3/2018, EU General data protection Regulation 2016/679 and Law 14/2007 for biomedical research). All participants signed an informed consent form after receiving information of the study and prior to obtaining biological samples. The biological material obtained was kept at ICO and processed under the appropriate measures to preserve the confidentiality of the results and data.

**Provenance and peer review** Not commissioned; externally peer reviewed.

**Data availability statement** Data are available on reasonable request.

**ORCID iDs**
Anna Saura-Lázaro http://orcid.org/0000-0001-9742-2725
Yolanda Benavente-Moreno http://orcid.org/0000-0003-1422-4614
Delphine Casabonne http://orcid.org/0000-0002-7874-3707
Sandra Cabrera http://orcid.org/0000-0003-3013-8812
Adaia Albasanz-Puig http://orcid.org/0000-0001-9852-5574

Carlota Gudiol http://orcid.org/0000-0003-3095-4422
Carles Muñoz-Montplet http://orcid.org/0000-0002-7324-8889
Àngel Plans http://orcid.org/0000-0003-3199-0361
Dolors Carnicer-Pont http://orcid.org/0000-0002-3475-8704
Laia Alemany http://orcid.org/0000-0003-0945-6015
Esteve Fernández http://orcid.org/0000-0003-4239-723X

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
