## [Reviewer comments · BMJ Open]

ARTICLE DETAILS

TITLE (PROVISIONAL)	COVID-19 among workers of a Comprehensive Cancer Center between first and second epidemic waves (2020): a seroprevalence study in Catalonia, Spain.
AUTHORS	Peremiquel-Trillas, Paul; Saura-Lazaro, Anna; Benavente Moreno, Yolanda; Casabonne, Delphine; Loureiro, Eva; Cabrera, Sandra; Duran, Angela; Garrote, Lidia; Brao, Immaculada; Trelis, Jordi; Galán, Maica; Soler, Francesc; Julia, Joaquim; Cortasa, Dolça; Domínguez, Maria Ángeles; Albasanz-Puig, Adaia; Gudiol, Carlota; Ramírez-Tarruella, Dolors; Muniesa, Joan; Rivas, Juan; Muñoz-Montplet, Carles; Sedano, Ana; Plans, Àngel; Calvo-Cerrada, Beatriz; Calle, Candela; Clopés, Ana; Carnicer-Pont, Dolors; Alemany, L.; Fernandez, Esteve

VERSION 1 – REVIEW

REVIEWER	Varona, Jose HM Hospitales, Internal Medicine
REVIEW RETURNED	12-Sep-2021

GENERAL COMMENTS	Interesting study that provides data of interest in the environment of healthcare workers specifically in oncology There are several points that need to be improved in the manuscript In the Abstract in the conclusions section the word ICO must be explained An explanation is necessary that it occurred with 35% of the workers who did not agree to participate in the study, since this may constitute a significant selection bias It is recommended to categorize the professional category based on the degree of exposure to COVID-19: • high risk exposure, including those workers who carry out their activity in a clinical environment and have prolonged direct contact with patients (eg, nurse, doctor, physiotherapist, porter, etc)• moderate risk exposure, including those who work in a clinical environment and have non-intense/no patient contact, but are potentially at higher risk of nosocomial exposure (eg, domestic and laboratory staff)• low risk exposure, which included those staff who work in a non-clinical environment and have minimal/no patient contact (eg, office staff/administrative, information technology, secretarial, clerical).
---

	It is interesting to perform the differential seroprevalence analysis based on these three categories of exposure to COVID-19 We recommend shortening the discussion as it is excessively long. One of the main weaknesses is the lack of updating of the bibliographic references. It is evident that the bibliography search should be reviewed and updated, because the manuscript has not considered one of the main seroprevalence studies in health workers that evaluate more than 6000 subjects (reference: Varona JF, Madurga R, Peñalver F, et al. Seroprevalence of SARS-CoV-2 antibodies in over 6000 healthcare workers in Spain. Int J Epidemiol. 2021; 50 (2): 400-409. doi: 10.1093 / ije / dyaa277) Bibliographic references are often poorly cited
--	--

REVIEWER	Varona, Jose HM Hospitales, Internal Medicine
REVIEW RETURNED	19-Feb-2022

GENERAL COMMENTS	Now, manuscript is OK
-----------------------

VERSION 1 – AUTHOR RESPONSE

Thanks for the opportunity to review again the manuscript.

As suggested we have :

1. Included Carlota Gudiol in the contributorship statement.
2. Converted to PDF the suppl file
3. Update the old title (that was changed as per the reviewers' suggestion) in the system
4. Revised the references and included number 37 in its place (don't know how it disappeared!).